# Coenzyme A in Brain Biology and Neurodegeneration

**DOI:** 10.3390/biomedicines14010069

**Published:** 2025-12-29

**Authors:** Dejun Zhang, Charlie Brett, Jason Cho, Tammaryn Lashley, Ivan Gout

**Affiliations:** 1Department of Structural and Molecular Biology, University College London, London WC1E 6BT, UK; dejun.zhang.20@ucl.ac.uk (D.Z.); charlie.brett.21@ucl.ac.uk (C.B.); jason.cho.22@ucl.ac.uk (J.C.); 2Department of Neurodegenerative Disease, UCL Queen Square Institute of Neurology, London WC1N 3BG, UK; t.lashley@ucl.ac.uk; 3Institute of Molecular Biology and Genetics, National Academy of Sciences of Ukraine, 03143 Kyiv, Ukraine

**Keywords:** Coenzyme A, neurodegeneration, redox regulation

## Abstract

Coenzyme A (CoA) biology has been extensively studied in health and disease due to the central role of CoA in numerous metabolic and signalling processes. CoA is essential for all living organisms, and its biosynthesis and homeostasis are tightly regulated by nutrient availability, mitogenic stimuli, and stress signals. Disruptions in CoA biosynthesis, caused by inborn mutations in genes encoding enzymes of the CoA biosynthetic pathway (such as *PANK2* and *CoASy*), lead to neurodegeneration, indicating the critical role of CoA/CoA thioesters in the function and viability of neuronal cells. The molecular mechanisms linking CoA deficiency to neurodegeneration remain unknown, but recent studies have highlighted the involvement of disrupted metabolism and redox homeostasis. The antioxidant function of CoA, mediated by protein CoAlation, has recently emerged as a novel and important mechanism of redox regulation. This review highlights well-established principles of CoA in neuronal metabolism and summarises recent advances in our understanding of its role in adaptive responses to oxidative and metabolic stress. The identification of enzymes involved in the CoAlation/deCoAlation cycle, together with the development of novel analytical tools and methodologies, may provide new insights into the discovery of more effective diagnostic and therapeutic approaches for targeting neurodegenerative diseases.

## 1. Introduction

The human brain is an energy-demanding organ, consuming vast amounts of fuel to power our every thought, sensation, and action. Although it accounts for only approximately 2% of our body weight, it consumes roughly 20% of an individual’s energy expenditure [1]. Remarkably, the brain synthesises approximately 5.7 kg of ATP daily, with about 70% utilised by neurons and the remainder by glial cells, including astrocytes, oligodendrocytes, and microglia [2,3,4].

Different brain cell types exhibit distinct metabolic profiles, with neurons requiring the most ATP to sustain their specialised functions. To meet these high energy demands, neurons rely heavily on the continuous uptake and breakdown of nutrients, including glucose, lactate, and ketone bodies. These nutrients are supplied directly from nearby capillaries or via the glial cells [5]. Neurons metabolise these fuels through glycolysis, oxidative phosphorylation, and the oxidation of lactate and ketone bodies to generate ATP. Neurons are especially sensitive to imbalances in energy production, with prolonged energy crisis rendering them susceptible to dysregulated synaptic function and neuronal activity. Mitochondria are the powerhouses of neurons, and their dysfunction, leading to deficits in cellular energy production is recognised as a major driver in the onset and progression of neurodegenerative diseases.

CoA is an essential metabolic cofactor that is required for the survival of all living cells [6,7,8,9]. It plays pivotal roles in cellular processes such as ATP biogenesis, neurotransmitter synthesis, and the regulation of gene expression [9,10]. Under physiological conditions, CoA functions as a key cellular metabolite, using its highly reactive thiol group to form metabolically active CoA thioesters [8,9,10]. Through this mechanism, CoA facilitates the delivery of energy-rich carboxylic acids to different subcellular compartments, priming them for entry into specific metabolic pathways, including energy production [6,11].

Recent findings suggest that under oxidative or metabolic stress, CoA may shift from its traditional metabolic role to act as a cellular antioxidant, protecting cells from irreversible oxidative damage [9,10,12]. CoA accomplishes this novel function by covalently modifying redox-sensitive cysteine residues on cellular proteins, a process now termed protein CoAlation. Recent advances in research tools and methodologies have enabled the identification of several thousand CoAlated proteins (the “CoAlome”) in both eukaryotic and prokaryotic cells exposed to oxidative stress [13,14]. Bioinformatic analysis of all CoAlated proteins in the CoAlome database revealed that over 60% of them are implicated in metabolic processes [13,14]. Protein CoAlation has been shown to induce significant conformational changes, modulate the activity, and alter the subcellular localisation of CoAlated proteins, along with protecting them from irreversible sulfhydryl overoxidation [15,16,17].

Dysregulation of CoA biosynthesis due to inborn mutations in pantothenate kinase 2 (*PANK2*) and CoA synthase (*CoASy*) is associated with neurodegeneration with brain iron accumulation (NBIA) [18,19]. These mutations disrupt mitochondrial energy metabolism, lipid synthesis, and neuronal maintenance, leading to iron accumulation in the basal ganglia and neurodegeneration.

Hallmarks of neurodegeneration include neuronal cell death, protein aggregation, inflammation, dysregulated gene expression, and neuronal network defects, alongside altered energy homeostasis [20]. Emerging research has shown that many of these pathological features are influenced by disruptions in cellular metabolism and oxidative stress, two processes in which CoA plays a central role [14,21]. Dysregulated energy metabolism and oxidative stress can contribute to a critical energy crisis in neurons, a key factor driving the progression of neurodegeneration. Accordingly, CoA’s metabolic and antioxidant properties might harness neuroprotective benefits in mitigating such pathologies.

## 2. CoA in Brain Metabolism

Under normal conditions, neurons generate ~7% of their ATP through glycolysis, while the majority of ATP (~93%) is produced via mitochondrial oxidative phosphorylation [3]. Metabolic processes that are involved in neuronal energy production are shown in Figure 1. CoA is not directly involved in glycolysis, since glycolytic enzymes do not use it as a cofactor or substrate. The final product of glycolysis, pyruvate, is transported into the mitochondria by mitochondrial pyruvate carrier 2/SLC54A [22]. In the mitochondrial matrix, pyruvate is oxidised to acetyl-CoA by the pyruvate dehydrogenase complex (PDC) or carboxylated to oxaloacetate (OAA) by pyruvate carboxylase [23]. Additionally, pyruvate can also be produced from (a) alanine by alanine transaminase; (b) lactate by lactate dehydrogenase B; or (c) malate by malic enzymes.

Outside the mitochondria, acetyl-CoA is also produced in the cytosol by acetyl-CoA synthetases, which catalyse the ligation of CoA to free acetate using ATP [24]. Depending on cellular demands, the generated acetyl-CoA primarily enters the tricarboxylic acid (TCA) cycle to drive mitochondrial ATP production. Additionally, acetyl-CoA can also be utilised for the synthesis of fatty acids, cholesterol, and acetylcholine when its derived intermediates, such as citrate, are exported from the mitochondria [25]. In the cytosol, these intermediates are converted back into acetyl-CoA and used in specific biosynthetic pathways. The resulting ATP, fatty acids, cholesterol, and acetylcholine subsequently support synaptic transmission, myelin biosynthesis, and the restoration of the resting membrane potential (Figure 1).

The production of neuronal ATP is almost entirely dependent on the availability of CoA, an essential metabolic cofactor. CoA participates in nearly all metabolic and signalling pathways that drive neuronal energy production and expenditure. Within neurons, nutrients are catabolised into energy-rich carboxylic acids, which are then conjugated to CoA to produce metabolically active thioesters such as acetyl-CoA, malonyl-CoA, and HMG-CoA. These thioesters act as molecular vehicles, shuttling carboxylic acids to specific subcellular compartments to support energy metabolism and cellular biosynthetic processes.

Among these derivatives, acetyl-CoA is the best-studied thioester, playing key roles in ATP biosynthesis, neurotransmitter synthesis, and the regulation of gene expression. For instance, acetyl-CoA enters the TCA cycle and fuels a series of redox reactions that generate key electron carriers, such as NADH and FADH_2_. These carriers subsequently donate electrons to the mitochondrial electron transport chain (ETC), driving ATP production via oxidative phosphorylation [24]. Acetyl-CoA also donates its acetyl group to choline to produce the neurotransmitter acetylcholine, in a reaction catalysed by choline acetyltransferase (ChAT) [25]. The function of cholinergic neurons critically depends on acetylcholine production, and reduced ChAT activity has been closely associated with Alzheimer’s Disease (AD) progression [26]. Furthermore, acetyl-CoA regulates gene expression, serving as a substrate for histone acetyltransferases, which acetylate lysine residues on histone proteins. Nuclear acetyl-CoA synthetase 2 (ACSS2) directly regulates histone acetylation in neurons, and inhibition of ACSS2 expression in mice leads to impaired long-term spatial memory [27,28].

Beyond its own metabolic and signalling roles, acetyl-CoA serves as a precursor for the synthesis of many other important CoA thioesters. For example, malonyl-CoA, formed via the carboxylation of acetyl-CoA, is a key substrate for fatty acid synthesis in neurons and glia. This pathway is essential for myelination by oligodendrocytes, which enhances the speed and fidelity of action potential conduction along axons. Another key derivative, HMG-CoA, participates in cholesterol biosynthesis. Cholesterol is a major component of myelin, where it stabilises the myelin sheath and protects axons from electrical damage [29]. In addition, HMG-CoA is involved in the biosynthesis of β-Hydroxybutyrate (BHB), a vital alternative energy fuel that can be shuttled to neurons for ATP production during periods of starvation/ketogenesis. Altogether, CoA and its thioester derivatives play a central role in brain metabolism and functionality, serving as a bridge between nutrient catabolism and ATP production.

While the high metabolic rate of neurons is essential for supporting their specialised functions, it comes with a significant drawback: elevated production of reactive oxygen species (ROS) as a byproduct of intensified mitochondrial respiration. In addition to mitochondria (mainly by the electron transport chain), ROS are also produced at the plasma membrane (primarily by NADPH oxidases), the endoplasmic reticulum (through oxidative protein folding and the formation of disulfide bonds, mediated by disulfide isomerases and oxidoreductases). Key examples of endogenously produced ROS include superoxide anion (O_2_^−^), hydrogen peroxide (H_2_O_2_), hydroxyl radical (⋅OH), organic peroxides (e.g., lipid peroxides), peroxynitrite (ONOO^−^), hypochlorous acid (HOClHOCl), etc. These molecules primarily arise from electron leakage in the mitochondrial ETC. At physiological levels, ROS has beneficial signaling roles, regulating cytoskeletal dynamics, synaptic plasticity, and neuronal differentiation [30]. However, excessive ROS production can overwhelm the antioxidant defence system, resulting in oxidative stress. This redox imbalance has been extensively implicated in the pathogenesis of several neurodegenerations.

## 3. Oxidative Stress and Antioxidant Defence in Neuronal System

This sensitivity to oxidative stress is exacerbated by the fact that mitochondria, the engines of neuronal energy production, are also the primary source of reactive oxygen species. Under normal physiological conditions, ROS serve as signalling roles; however, excessive production overwhelms the antioxidant defence system, leading to oxidative stress. This imbalance has been extensively implicated in the pathogenesis of several neurodegenerative pathologies.

The primary contributors to oxidative stress are ROS, though reactive nitrogen species (RNS) also play a role. ROS originate from various intracellular sources (Figure 2), amongst them, the mitochondria are the predominant producers, accounting for approximately 90% of ROS in the body. This is particularly significant in neurons, where mitochondria are abundant in both cell bodies and synaptic terminals to meet the high metabolic demands of neurotransmission [31,32]. Within the mitochondrial ETC, complexes I and III are the main sites of ROS generation. Electrons leak from reduced flavin mononucleotide at complex I and from the semiquinone anion (^•^Q^−^) in the Q-cycle at complex III and are transferred to oxygen to form the superoxide radical (O_2_^•−^) [33,34].

Neurotransmitters themselves can also contribute to oxidative stress. For instance, glutamate, a major excitatory neurotransmitter implicated in AD and Huntington’s disease [35,36]. They can lead to overactivation of N-methyl-D-aspartate glutamate receptors, promoting calcium influx and subsequent mitochondrial dysfunction, resulting in increased mitochondrial ROS production [37]. Similarly, dopamine is inherently unstable and must be tightly regulated. Vesicular monoamine transporter 2 (VMAT2) sequesters cytosolic dopamine into synaptic vesicles. In VMAT2-deficient models, unsequestered dopamine undergoes oxidation in the cytosol, forming reactive dopamine-o-quinone, which can damage proteins and induce nigrostriatal neurodegeneration, a mechanism potentially contributing to Parkinson’s disease (PD) pathology [38].

Other subcellular compartments also contribute significantly to oxidative species production. In peroxisomes, β-oxidation of fatty acids generates hydrogen peroxide (H_2_O_2_) via acyl-CoA oxidase activity. Increased production H_2_O_2_ in neurons has been shown to activate amyloid precursor protein processing through beta-secretase, resulting in elevated amyloid-beta (Aβ) production [39]. Aβ in turn promotes further ROS formation from cholesterol and dopamine by complexing with a redox-active Cu^2+^ ion, creating a self-perpetuating cycle of oxidative damage and Aβ pathology, leading to AD [40]. Another enzyme called xanthine oxidase, involved in purine catabolism, produces O_2_^•−^ [41]. In the endoplasmic reticulum, cytochrome P450-dependent monooxygenases and Ero1p are key sources of ROS [42,43]. In lysosomes, nitric oxide synthases catalyse the oxidation of L-arginine to produce nitric oxide (NO^•^) and L-citrulline, contributing to RNS generation, while plasma membrane-bound NADPH oxidases produce O_2_^•−^ [44,45]. Additionally, exogenous factors like ionising radiation, alcohol, tobacco smoke, and heavy metals can also contribute to ROS and RNS formation [34].

Although physiological levels of ROS and RNS are crucial for redox signalling and synaptic plasticity, such as the recruitment of glutamate receptors to synapses [46]. However, excessive production of these species can overwhelm antioxidant defences and damage cellular components (Figure 2). 

Proteins are a major target for oxidative damage, especially for sulphur-containing amino acids. Cysteine, containing a side chain thiol group (-SH), can be reversibly oxidised into sulfenic acid (-SOH), sulfinic acid (-SO_2_H), and then irreversibly to sulfonic acid (-SO_3_H), changing its chemical properties and altering protein structure [47]. Reactive species such as dopamine-o-quinone can also form adducts with cysteines, damaging synaptic proteins [38]. Backbone cleavage may occur via ROS-mediated alkoxyl radical formation at α-carbons, leading to fragments like diamides and isocyanates or amides and ketoacyls [47,48]. Oxidative components of neurotoxins have been found to affect synaptic membrane protein specifically, affecting surface aromatic residues, leading to neuronal cell death, and links to neurodegeneration [49].

Polyunsaturated fatty acids (PUFAs) are highly concentrated in the brain and play crucial roles in maintaining neuronal structure, function, and health. PUFAs in membranes are targets for lipid peroxidation, initiated by increased production of •OH, leading to the accumulation oflipid peroxyl radicals. They attack adjacent fatty acids, creating lipid hydroperoxides (LOOH) and new radicals, propagating the chain reaction. Termination occurs when two radicals combine, but not before extensive membrane disruption [50]. This process is especially pronounced in neuronal membranes that are rich in PUFAs, which precedes and accompanies Aβ accumulation in AD models, fuelling deleterious cascades that impair membrane integrity and cellular functions [51]. •OH also attacks purine bases in DNA, producing 8-hydroxy-2′-deoxyguanosine (8-OHdG) lesions that can induce G:C to T:A transversion mutations and form abasic sites. This oxidative damage extends to both nuclear and mitochondrial DNA and RNA in AD, with mitochondrial DNA being especially susceptible due to its proximity to the ETC, contributing to mitochondrial dysfunction and energy deficits crucial to neuronal survival [52].

To confer protection against oxidative stress, cells rely on a robust antioxidant defense system (Figure 3). This system is composed of both enzymatic and non-enzymatic antioxidants that work together to eliminate a diverse range of ROS [53]. Enzymatic antioxidants include superoxide dismutase (SOD), catalase (Cat), and glutathione peroxidase (GPx), which constitute the first line of defense against ROS [54].

SOD plays a critical role in converting O_2_^•−^ into H_2_O_2_, the resulting H_2_O_2_ can be detoxified directly into water and oxygen by catalase [55]. However, mutations in *SOD* causes amyotrophic lateral sclerosis [56]. Alternatively, GPx reduces H_2_O_2_ to water through a process involving the oxidation of glutathione (GSH) to glutathione disulfide (GSSG) [57]. Other major enzymatic antioxidants include the thioredoxin (Trx) and peroxiredoxin (Prx) redox systems, which collectively contribute to H_2_O_2_ detoxification by reducing it to water [58]. If not fully reduced, H_2_O_2_ can initiate a cascade of damaging events that promote cell death. For instance, in the presence of ferrous iron (Fe^2+^), H_2_O_2_ is converted into the highly reactive hydroxyl radical (•OH) via the Fenton reaction, which can damage cellular components and contribute to ferroptosis [59].

While much of the antioxidant defense system relies on enzymatic mechanisms, ROS can also be neutralised through non-enzymatic pathways involving vitamins. For instance, vitamin C (ascorbate) reacts with several ROS, forming a stable and relatively unreactive ascorbyl radical, thereby protecting cells from ROS-induced damage [60]. Ascorbate is one of the most abundant antioxidants in the brain and undergoes continuous recycling between neurons and glia [61]. Upon oxidation by ROS, it is released from neurons, reduced back to ascorbate within glial cells, and subsequently transported into neurons, thereby maintaining high intracellular levels to counteract oxidative stress [61,62].

Additionally, vitamin C enhances the antioxidant capacity of vitamin E, another important cellular antioxidant [63]. Vitamin E scavenges peroxyl radicals and plays a critical role in preventing lipid peroxidation [64]. Other non-enzymatic antioxidants, such as GSH, selenium, uric acid, metal-binding proteins, and various metabolites, also contribute to ROS detoxification [65]. Notably, GSH is a particularly important antioxidant that not only protects proteins from irreversible oxidative damage through covalent attachment; but also contributes to the activation of several pro-survival signalling pathways in response to oxidative stress. This protective modification, known as S-glutathionylation, can occur spontaneously or be catalysed by glutathione S-transferases, such as GSTpi, which facilitates the enzymatic transfer of GSH to oxidised protein thiol groups. Like GSH, CoA is another important low-molecular-weight thiol involved in antioxidant defence and redox regulation, which protects proteins from oxidative damage through a novel PTM termed CoAlation.

## 4. Antioxidant Functions of CoA in the Brain

CoA was discovered in the middle of the last century, and since that time research has mainly focused on the functions of CoA and its thioesters in cellular metabolism and the regulation of gene expression (Figure 4). Advances in exploring the involvement of CoA in antioxidant defence as a low molecular weight (LMW) thiol were previously hindered by the lack of specific analytical tools and methodologies. Progress was made when a highly specific anti-CoA monoclonal antibody (mAb) was developed and showed the ability to detect CoA-modified proteins by various immunological assays under oxidative or metabolic stress [66]. To our knowledge, there are no commercially available anti-CoA mAbs, even though CoA was discovered more than seventy years ago. The availability of a highly specific anti-CoA antibody (1F10) enabled the first experimental demonstration of CoA’s antioxidant function and protein CoAlation in cell lines, tissues, and model organisms exposed to oxidising agents or metabolic stress.

These studies revealed extensive covalent modification of protein thiols by CoA in primary cells and rat tissues [14]. However, protein CoAlation in established cell lines was much weaker under oxidative stress. These observed differences reflected the lower levels of CoA in established cell lines compared with primary cells, rat heart, and liver tissues [14]. Genetic upregulation of CoA biosynthesis in HEK293 cells through stable expression of Pank1β resulted in a sixfold increase in CoA levels and significantly elevated protein CoAlation under oxidative stress [14]. The widespread and reversible nature of protein CoAlation was then demonstrated in model organisms and mammalian cells [13,14,67]. The ability of our anti-CoA mAb to pull down CoAlated tryptic peptides and the knowledge of how to prepare them for mass spectrometry (MS) analysis facilitated the development of a reliable MS-based methodology for the detection of CoAlated proteins [14]. To date, this methodology has allowed the identification and characterisation of over 2300 CoAlated proteins in eukaryotic cells/tissues and prokaryotes [13,14]. Bioinformatic analysis showed that most of them (over 60%) are involved in metabolic processes, followed by proteins implicated in translation and stress response. The identity of CoAlated proteins was a prerequisite for establishing efficient in vitro CoAlation/deCoAlation assays. These analytical advances and the use of biochemical, biophysical, crystallographic, cell and molecular biology approaches showed that CoAlation can regulate the activity, subcellular localisation, and conformation of modified proteins [17,21,68,69,70]. While CoAlation mostly inhibits the function of modified proteins through covalent modification of catalytically important cysteine residues, a recent study demonstrated that CoA binding can promote allosteric activation of modified proteins. For instance, CoAlation of the antioxidant enzyme thioredoxin reductase 2 (TXNRD2), was shown to enhance its TXNRD2 activity, thereby protecting cells against mitochondrial lipid peroxidation and oxidative damage [70]. This finding suggests that CoAlation is potentially involved in redox signalling and activating proteins that drive the cellular antioxidant response (Figure 5). Like other LMW thiols (GSH, BSH and MSH), CoA was shown to protect oxidised cysteine residues from irreversible overoxidation [13].

By analogy to glutathione, a putative CoA redox cycle has been recently proposed. It may include the following enzymes: (a) CoA transferase, promoting protein CoAlation, (b) CoAredoxin, facilitating protein deCoAlation, (c) CoA-dependent peroxidases, eliminating peroxide radicals, and (d) CoA disulfide reductase (CoADR), catalysing the reduction in CoA disulfides. To date, CoADR was identified in Gram+ bacteria and archaea, having been shown to catalyse the reduction in CoA disulfides to CoASH [71]. However, the identity of eukaryotic CoADR is unknown and remains the subject of ongoing research. Since the developed anti-CoA mAb was also found to detect CoAlated proteins by immunohistochemistry (IHC) and immunofluorescence (IF), these methodologies were employed to examine protein CoAlation patterns in pathologies associated with oxidative stress, including neurodegeneration [21]. Another incentive to perform this study was related to the fact that inborn mutations in genes involved in the CoA biosynthetic pathway, such as *Pank2* and *CoASy*, lead to NBIA [18,19]. The anti-CoA IHC analysis of post-mortem brain samples of various neurodegenerative pathologies revealed significantly increased immunoreactivity in samples from NBIA, PD, and AD, when compared to matched controls [21]. In this analysis, extensive anti-CoA immunoreactivity has been detected in AD brain samples with predominant localisation in structures resembling neurofibrillary tangles (NFTs). NFTs represent a neuropathological hallmark of AD and consist primarily of abnormally hyperphosphorylated and aggregated Tau proteins. Double immunostaining with anti-CoA and anti-Tau antibodies showed significant overlapping of immunoreactive signals and recombinant Tau was found to be CoAlated in vitro and in vivo. The site of Tau CoAlation was mapped to its microtubule binding region and in vitro CoAlation prevented H_2_O_2_-induced dimerisation through blocking intermolecular disulfide bond formation [21]. Genetic disruption of the CoA biosynthetic pathway has been investigated in various model organisms, including yeast, zebrafish, flies, and mice. Yeast models were first to be developed and demonstrated the essential nature of CoA for cell viability, mitochondrial function, iron accumulation, and increased sensitivity to oxidative stress [72].

Disrupted CoA biosynthesis has also been investigated in zebrafish, focusing primarily on Parkin and CoASy. Knockdown of the *Pank2* gene in zebrafish embryos results in decreased CoA levels, mitochondrial disorganisation, increased oxidative stress, developmental and locomotor abnormalities [73]. The abrogation of CoASy expression in zebrafish by a specific morpholino led to strong reductions in CoA levels, decreased bone morphogenic protein signalling, perturbated neurogenesis, and high lethality [74]. Multiple genetic models of disrupted CoA biosynthesis in *D. melanogaster* have also been developed and were highly informative for studying the effects of CoA deficiency, being the best-characterised in vivo systems linking metabolic stress to neurodegeneration. Established models, especially the *fumble (dPank/fbl)* mutants, linked impaired de novo CoA biosynthesis to mitochondrial defects, abnormal lipid metabolism, DNA damage, neurodegeneration and shortened lifespan [75,76].

Multiple mouse models have been created for studying CoA deficiencies by targeting CoA biosynthetic enzymes. They range from complete to tissue-specific knockouts and aim to reproduce biochemical and clinical features of human CoA biosynthesis disorders. The disruption of CoA biosynthesis in mice has been focused primarily on *Pank* and *CoASy* genes, since inborn mutations in human *Pank2* and *CoASy* were shown to lead to neurodegeneration [19]. Mice with total *Pank2* knockouts were initially developed in 2005 and manifested retinal degeneration, male infertility, ~20% decreases in weight, but no signs of brain iron accumulation and basal ganglia neurodegeneration [77].

Initially, the absence of a neurodegenerative phenotype in established mouse models has been explained by the different cellular localization of the human and mouse Pank2, and compensatory expression of other Pank isoforms. However, detailed analysis of Pank2 expression in the brain of wild type mice indicated that Pank2 is localised in the mitochondrial fraction but absent in the samples from *Pank2*^−/−^ mice [78]. This study also showed that neuron-specific knockout of *Pank2* in mice showed moderate decrease in CoA levels (~15%), mitochondrial abnormalities and reduced motor coordination, but no iron accumulation and gross neurological dysfunction [78]. At present, it remains unclear why Pank2 deficient mice do not develop neurodegeneration, despite the fact that mutations in human *Pank2* result in severe neurodegenerative phenotype. Studies from different laboratories revealed that genetic background or experimental setups are not associated with the lack of PKAN signs in established Pank2 mouse models.

*Pank1*-deficient mice were also generated and showed metabolic alterations rather than neurological deficits [79]. The phenotype is characterised by mild fasting hypoglycaemia, improved insulin sensitivity, and reduction in β-oxidation. Observed increases in carbohydrate utilisation, which coincides with reduced fatty acid and amino acid metabolism, suggests the role of CoA in metabolic reprogramming. The level of CoA was shown to be reduced in the liver (~40% lower), but not in other tissues which is possibly due to compensation by the PanK2 and/or PanK3 isoform. Combined PANK1/2 deficiency in mice produces the clearest CoA-deficiency and neurodegenerative phenotype with biochemical parallels to NBIA. It leads to early postnatal death, significantly reduced CoA levels (especially in liver and brain), severe hypometabolism, and neuronal deficits.

Total *CoASy* knockout results in embryonic lethality, confirming absolute requirement for CoA synthase [80]. To bypass this phenotype and to explore CNS-specific effects of CoA deficiency, conditional neuron-specific deletion of *CoASy* in mice was developed and showed altered iron homeostasis and mitochondrial function, severe early-onset neurological phenotypes, locomotor abnormalities, dystonia-like movements, growth arrest, and early postnatal death [80]. Surprisingly, total brain CoA levels were unchanged, and no signs of neurodegeneration were observed.

Building on these findings, the localisation and distribution of vitamin B5 (D-pantothenic acid), the essential precursor of CoA, has been investigated in the human brain. Pantothenate immunoreactivity was widespread in the caudate, putamen and cerebellum, Pantothenate-staining intensity correlated closely with the distribution of myelinated structures as identified by Nissl staining in both brain regions [81]. The presence of substantive pantothenate levels throughout the normal brain is consistent with the existence of pantothenate stores that could well participate in localised metabolic processes, such as those entailed in myelin synthesis [81]. In dementia, however, pantothenic acid levels were significantly decreased in six of the ten investigated brain regions: the pons, substantia nigra, motor cortex, middle temporal gyrus, primary visual cortex, and hippocampus [82].

Decreased CoA levels in the brain have been associated with oxidative stress, mitochondrial dysfunction, and neurodegeneration. Consequently, small-molecule drugs and CoA intermediates that modulate the CoA biosynthetic pathway have been investigated for their neuroprotective potential in both cellular and animal models.

One of the most extensively studied small-molecule modulators of CoA biosynthesis is PZ-2891, an allosteric activator of Pank. PZ-2891 binds Pank at the pantothenate site and stabilizes the active conformation of Pank. This allosteric mode of PZ-2891 binding to Pank alleviates feedback inhibition by CoA and Acetyl-CoA and increases CoA biosynthesis and its intracellular level. In a mouse model of PKAN, treatment with PZ-2891 improved survival, motor function, and body weight [83]. In a separate study, PZ-2891 showed promising pharmacokinetic and pharmacodynamic properties in the treatment of Aβ1-42 induced AD mouse models with effective brain-blood barrier penetration ratio of 0.59 [84]. More recently, an optimised derivative of PZ-2891, known as BBP-671, has been developed. Like PZ-2891, BBP-671 locks Pank in an active conformation, thereby suppressing feedback inhibition by CoA and acetyl-CoA. It also exhibits high stability and membrane permeability, with evidence indicating efficient penetration across the blood–brain barrier [85]. In PKAN mouse models, oral administration of BBP-671 significantly elevated brain CoA levels and improved both locomotor performance and survival rates [85]. Currently, the outcomes of clinical studies are only available for BBP-671. Phase I clinical trial (NCT04836494) has shown that administration of BBP-671 results in limited adverse events, effective at crossing brain-blood barrier and raising plasma acetyl-CoA levels [86]. However, the long-term safety of BBP-671 is still under evaluation in ongoing clinical trials, and its effectiveness in treating PKAN at various stages of the disease and propionic acidemia (a rare genetic metabolic disorder associated with deficient CoA metabolism).

Beyond targeting PANK, nutritional supplementation with various CoA intermediates has also demonstrated neuroprotective effects. One example is pantothenic acid (pantothenate), the first intermediate in the CoA biosynthetic pathway. In a kainic acid-induced model of excitotoxicity, used to simulate features of AD and Huntington’s disease, pre-treatment with pantothenic acid (30–90 mg/kg) delayed seizure onset, improved memory retention, and attenuated neuronal damage, likely by mitigating oxidative stress [87].

Similarly, treatment with downstream intermediates in CoA biosynthesis, such as 4-phosphopantetheine (PPan), has been proposed as a viable PKAN treatment. In PKAN mouse models, oral administration of PPan (20 μg/g) for two weeks effectively restored brain CoA levels, normalised iron homeostasis, and improved dopamine metabolism, thereby correcting multiple disease features [88]. In line with this, pantethine (a dimeric form of pantetheine) has also shown benefits in a transgenic 5xFAD mouse model of AD. Long-term treatment reduced Aβ accumulation and mitigated behaviours associated with aggression and hyperactivity [89].

Notably, supplementation of PKAN fibroblasts with pantothenate or pantethine prevented iron accumulation, increased Pank2 expression, and reduced lipid peroxidation, indicating their potential to reverse pathological alterations in PKAN patients [90]. This was further supported by studies using induced pluripotent stem cell (iPSC)-derived neurons from PKAN patients, in which administration of CoA precursors or CoA itself restored mitochondrial function, reduced ROS formation, and rescued neuronal viability [91]. Collectively, these studies suggest that strategies aimed at improving CoA availability may offer promising therapeutic approaches for treating NDs.

## 5. Future Perspectives

The discovery of the antioxidant function of CoA is a relatively recent breakthrough that expanded the traditional view of CoA as a key metabolic integrator. It was facilitated by the development of specific research tools and methodologies capable of detecting and mapping protein CoAlation, a reversible PTM which protects protein thiols from oxidative damage and facilitates redox signalling. The molecular mechanisms governing the protein CoAlation cycle are not yet defined, and the identity of enzymes implicated in these processes remain to be determined. The analogy to the well-studied process of protein S-glutathionylation has prompted the identification and biochemical characterisation of the enzymes that mediate the antioxidant function of CoA and protein CoAlation. The pKa value of the CoA thiol group is higher than that of GSH (∼9.8 and 8.8, respectively). This means that under physiological pH CoA exists predominantly in its unreactive, protonated thiol form. The identification and characterisation of GSH-, BSH- and MSH S-transferases have established a parallel framework for the existence of an enzyme which can promote protein CoAlation (CoA S-transferase). Protein CoAlation is a reversible PTM [14], which is most likely mediated by CoAredoxins (there are five glutaredoxins in the human genome). The identity of enzymes capable of removing CoA from CoAlated proteins remains to be defined. To date, eight GSH-dependent peroxidases are known in mammalian cells/tissues, and it is anticipated that CoA-dependent peroxidase(s) will be identified and characterised in the next decade. Oxidised homo- and heterodimers of CoA, such as CoASSCoA and CoASSG, are formed during oxidative and metabolic stress in eukaryotic and prokaryotic cells. Their reduction to CoA and GSH is required for growth and proliferation. CoADR was identified and characterised in Gram+ bacteria and archaea more than 20 years ago, but not in eukaryotic cells. The progress in identifying and characterising mammalian reductases capable of reducing homo and heterodimers of CoA is expected in the next few years.

The development of specific anti-CoA mAb and MS-based methodology for the identification of CoAlated proteins was critical for uncovering the antioxidant function of CoA. Further advances in this direction are expected in the coming years, including (a) measuring the stoichiometry of CoAlation; (b) determining the ratio of CoA to CoA homo- and heterodisulfides; and (c) assessing dynamic cycle of protein CoAlation in cells and tissues. The progress in the development of these methodologies will be essential for functional studies defining the role of protein CoAlation in health and pathologies associated with oxidative stress, including neurodegeneration.

## Figures and Tables

**Figure 1 biomedicines-14-00069-f001:**
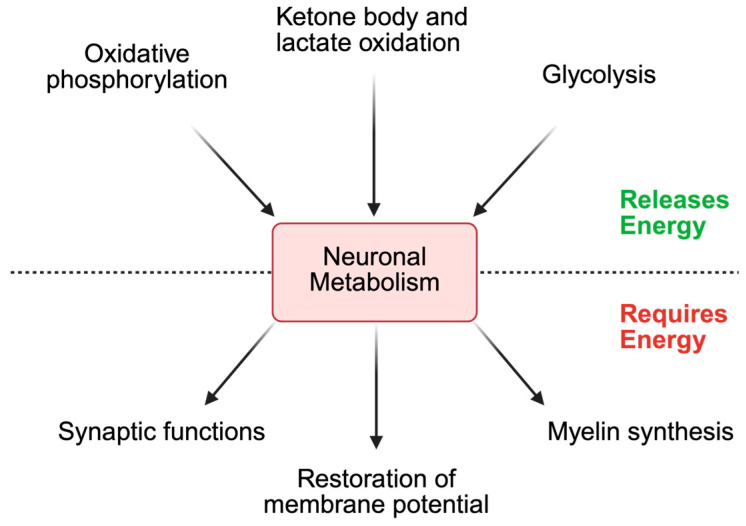
Metabolic processes underlying energy production and utilisation in neurons. Glycolysis, oxidative phosphorylation, and the oxidation of lactate and ketone bodies generates ATP. During glycolysis, glucose is converted to pyruvate, producing ATP. Lactate is converted to pyruvate, and ketone bodies such as β-hydroxybutyrate are oxidised to acetyl-CoA, allowing both intermediates to enter central metabolic pathways that yield ATP. This ATP is essential for restoring resting membrane potential, synthesising neurotransmitters, and producing myelin to improve signal transmission. Created in BioRender. Brett, C. (2025). https://BioRender.com/s04yyck.

**Figure 2 biomedicines-14-00069-f002:**
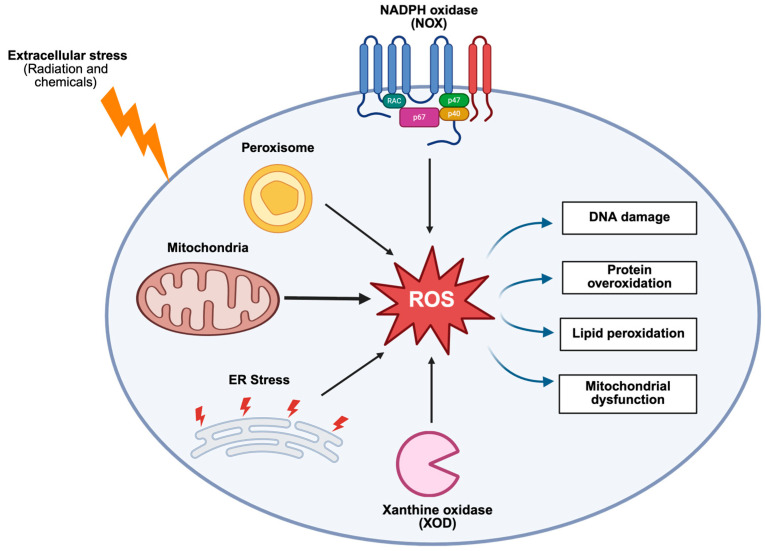
Sources and consequences of elevated reactive oxidative species in neurons. Schematic diagram illustrating both endogenous and exogenous sources of ROS, along with their cellular consequences. Among endogenous sources, mitochondria are a major contributor, as indicated by the increased arrow thickness in the diagram. Other internal sources include peroxisomes, the endoplasmic reticulum (ER), NADPH oxidase (NOX), and xanthine oxidase (XOD). Exogenous ROS sources include environmental stressors such as radiation and chemical exposure. Abnormal ROS levels are associated with multiple cellular consequences, including DNA damage, lipid peroxidation, protein overoxidation, and mitochondrial dysfunction. These processes disrupt cellular homeostasis and ultimately result in cell death. Created in BioRender. Brett, C. (2025). https://BioRender.com/i0xr6ra.

**Figure 3 biomedicines-14-00069-f003:**
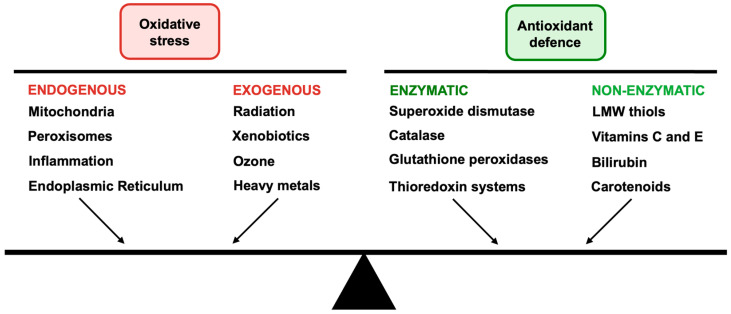
Schematic representation of the balance between oxidative stress and antioxidant defence. Left side shows the sources of oxidative stress, classified into endogenous sources from within the body, and exogenous sources originating from outside environments. The right side represents antioxidant defence mechanisms, categorised as enzymatic and non-enzymatic.

**Figure 4 biomedicines-14-00069-f004:**
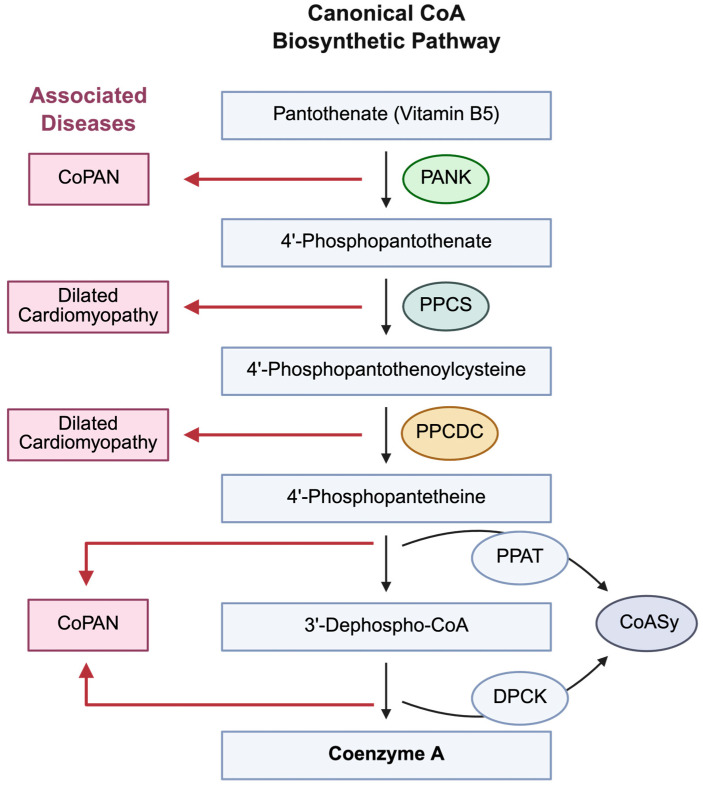
The CoA biosynthetic pathway in mammalian cells and associated pathologies. Schematic diagram of canonical CoA biosynthetic pathway in mammalian cells. Neurodegeneration is associated with inborn mutations in key CoA biosynthetic enzymes (Pank and CoASy). Abbreviations: Pantothenate kinase-associated neurodegeneration (PKAN); CoA synthase protein-associated neurodegeneration (CoPAN); Neurodegeneration with brain iron accumulation (NBIA). Pantothenate kinase (PANK); 4′-phosphopantothenoylcysteine synthase (PPCS); 4′-phosphopantothenoylcysteine decarboxylase (PPCDC); Phospho-panthethine transferase (PPAT); Dephospho CoA kinase (DPCK); CoA synthase (CoASy). Created in BioRender. Brett, C. (2025). https://BioRender.com/uzrb4jh.

**Figure 5 biomedicines-14-00069-f005:**
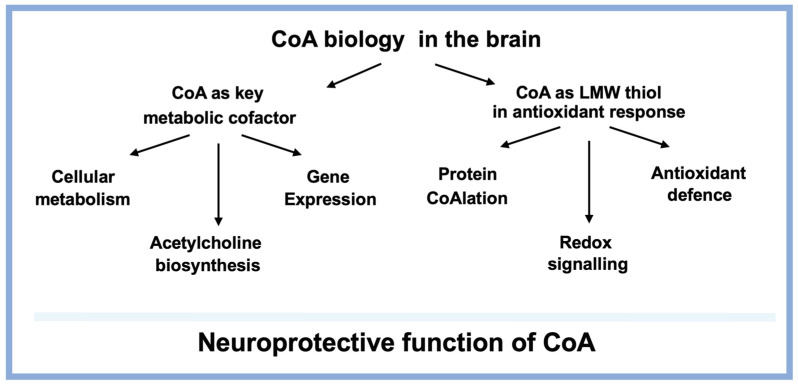
Representation of dual roles of CoA in brain biology. In its first role, CoA functions as a critical and metabolically active thioesters in the regulation of cellular metabolism, acetylcholine biosynthesis and gene expression. When cells/tissues are exposed to oxidative stress, CoA switches to maintain redox homeostasis through protein CoAlation, redox signalling and antioxidant defence.

## Data Availability

No new data were created or analyzed in this study. Data sharing is not applicable to this article.

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
