# Peer review of "Coenzyme A in Brain Biology and Neurodegeneration"

_biomedicines, 2025, doi:10.3390/biomedicines14010069_

Round 1
Reviewer 1 Report
Comments and Suggestions for Authors
The review article authored by Dejun Zhang et al. with the title "Coenzyme A in Brain Biology and Neurodegeneration" provides a comprehensive and timely synthesis of the recent understanding of the metabolism of coenzyme A and its multidirectional role in the neuronal function and emerging its significant in the modulation of pathology of neurodegenerative disease. The authors successfully bridged the gap between the traditional role of CoA as a central metabolic cofactor and its newly discovered function as a key factor in antioxidant defense via the CoA protein. This review is well-structured and effectively highlights key knowledge gaps and future research directions. The review is qualified to be published in biomedicine after modification of some points
- While the discussion of the CoAlization/de-CoAlization cycle is interesting, it is largely speculative. The authors could have clarified the established facts through the proposed hypotheses more clearly. For example, the existence of eukaryotic CoA disulfide reductase enzymes is currently unknown, a point that could have been emphasized more to temper expectations.
- The review presents promising preclinical data for several therapeutic compounds. However, a more critical discussion of the challenges of applying these findings to human patients would have been beneficial. Topics such as blood-brain barrier penetration, long-term safety, and the difficulty of treating advanced neurodegeneration could have been explored in more detail.
- The article provides an excellent summary of various studies, but it is less critical in its analysis. For example, when discussing the conflicting results from different mouse models in which the Pank2 gene was knocked out, a deeper analysis of the reasons for these differences (e.g., genetic background, experimental conditions) would have been highly valuable.
- In the description of the ROS origin, an elaborate explanation of the origin of all ROS is needed. The mentioned information is an overgeneralization that all listed ROS originate from the ETC.
Minor editing is required
Author Response
We would like to thank the Reviewers for their critical comments, and please find below our response.
Comment 1: While the discussion of the CoAlization/de-CoAlization cycle is interesting, it is largely speculative. The authors could have clarified the established facts through the proposed hypotheses more clearly. For example, the existence of eukaryotic CoA disulfide reductase enzymes is currently unknown, a point that could have been emphasized more to temper expectations.
Response 1: The enzymes which control the CoAlation/deCoAltion cycle have not been identified so far, and we agree with the Reviewer that it is important to distinguish hypothesis from facts. In the revised manuscript, we have clearly stated that the proposed involvement of CoA in redox regulation is speculative, and the work is currently in progress on the identification of enzymes implicated in this process is currently in progress. We have highlighted the edited portion; they are located between line 338-342.
Comment 2: The review presents promising preclinical data for several therapeutic compounds. However, a more critical discussion of the challenges of applying these findings to human patients would have been beneficial. Topics such as blood-brain barrier penetration, long-term safety, and the difficulty of treating advanced neurodegeneration could have been explored in more detail.
Response 2: We have added additional information regarding the effectiveness, safety, and brain-blood barrier penetration of PZ-2891 and BBP-671 in pre-clinical and clinical studies. We have highlighted the changes; they are located between line 430-450.
Comment 3: The article provides an excellent summary of various studies, but it is less critical in its analysis. For example, when discussing the conflicting results from different mouse models in which the Pank2 gene was knocked out, a deeper analysis of the reasons for these differences (e.g., genetic background, experimental conditions) would have been highly valuable.
Response 3: We have updated the manuscript by discussing conflicting findings produced from different KO and KI mouse models (p11). Furthermore, we have also stated that studies from different laboratories revealed no association between genetic background or experimental setups and the lack of PKAN signs in established Pank2 mouse models. We have highlighted the changes; they are located between line 386-397.
Comment 4. In the description of the ROS origin, an elaborate explanation of the origin of all ROS is needed. The mentioned information is an overgeneralization that all listed ROS originate from the ETC.
Response 4: In the revised manuscript, we have provided additional information about the production of ROS in various subcellular compartments, including mitochondria (specifically by the electron transport chain), the plasma membrane (primarily generated by NADPH oxidase (NOX) enzymes), the endoplasmic reticulum (through oxidative protein folding and the formation of disulfide bonds, mediated by disulfide isomerases and oxidoreductases). We have highlighted the changes; they are located between line 145-149.
Reviewer 2 Report
Comments and Suggestions for Authors
This is an interesting review and summary of the recent progress around the research of coenzyme A and its involvement in redox processes and antioxidant function with relevance to neurodegeneration. It is basically well written but some minor improvments would be necessary to make it more easy to digest.
My main concern is the awkward placement of the Figures. E.g. Figure 2 is first mentioned on page 4, but not shown until page 6. Figure 4 is first mentioned on page 6 but not shown until pages 10. There is a Figure 5 but I cannot find where it is mentioned in the text. So, pls edit the text/placement of the Figures so that they appear shortly after they are referred to.
The Figures are also very different when it comes to the style. E.g. Figure 1 uses huge fonts to deliver some messages in boxes. Similar boxes are used in Figure 3 and 4 but with completely different fonts and style.
Also, there are many complicated reactions and processes that could have some additional graphics, e.g. the enzymatic CoA redox cycle (described starting from line 311).
Author Response
We would like to thank the Reviewers for their critical comments, and please find below our response.
Comment 1: My main concern is the awkward placement of the Figures. E.g. Figure 2 is first mentioned on page 4, but not shown until page 6. Figure 4 is first mentioned on page 6 but not shown until pages 10. There is a Figure 5 but I cannot find where it is mentioned in the text. So, pls edit the text/placement of the Figures so that they appear shortly after they are referred to.
Response 1: We agree with Reviewer’s comment about the awkward placement of figures in the text. Therefore, the location of Figures 2 and 4 has been changed in the revised manuscript. Furthermore, the description of Figure 5 is in the text was misplaced due to a software error and this issue is now resolved.
Comment 2: The Figures are also very different when it comes to the style. E.g. Figure 1 uses huge fonts to deliver some messages in boxes. Similar boxes are used in Figure 3 and 4 but with completely different fonts and style.
Response 2: We agree with the Reviewer and change the font in all Figures to Arial. Furthermore, shades of red and green in Figures 3 and 5 correspond to those used in Figure 1. The size of figures in the revised manuscript was also adjusted.
Comment 3: Also, there are many complicated reactions and processes that could have some additional graphics, e.g. the enzymatic CoA redox cycle (described starting from line 311).
Response 3: As suggested, we have modified the description of complex reactions and processes to make it easier for readers to understand. There are already five Figures in this review, and we feel that having an additional figure might be inappropriate
Round 2
Reviewer 2 Report
Comments and Suggestions for Authors
The authors made a good job making the Graphics more uniform. There is however, some simple editing tricks that could improve it even further, e.g.
Figure 2 could be placed around line 183.
Figure 3 placed at current lines 242 or 252.
Figure 4 better shown before referring to Figure 5. E.g. by making a line-break at current line 289 and place it there. Or another similar adjustment of that paragraph.
The placement and style of figures become especially important when it comes to reading heavy text papers on-line. With a paper print-out this is usually a minor problem.
Author Response
We would like to thank the Reviewers for their critical comments, and please find below our response.
Comment 1: Figure 2 could be placed around line 183.
Response 1: We have moved Figure 2 to around line 183
Comment 2: Figure 3 placed at current lines 242 or 252.
Response 2: We have moved Figure 3 to line 250
Comment 3: Figure 4 better shown before referring to Figure 5. E.g. by making a line-break at current line 289 and place it there. Or another similar adjustment of that paragraph.
Response 3: We have introduced a line break at 290 and have placed figure 4 there.